# Priming exposures to lipopolysaccharides do not affect the induction of Polycomb target genes upon re-exposure

**Marco Geigges** [1], **Geethika Arekatla** [1], **Renato Paro** [1,2]*

**1** Department of Biosystems Science and Engineering, Epigenomics Group, ETH Zurich, Basel, Switzerland,
**2** Faculty of Science, University of Basel, Basel, Switzerland

\* renato.paro@bsse.ethz.ch

**Data Availability Statement:** All relevant data are within the paper and its Supporting Information files.

**Funding:** The research of MG, GA and RP was supported by the ETH Zurich (www.ethz.ch) and an

## Abstract

The Polycomb group (PcG) proteins are chromatin factors underlying the process of transcriptional memory to preserve developmental decisions and keep cellular identities. However, not only developmental signals need to be memorized and thus maintained during the life of an organism. For host protection against pathogens, also a memory of previous exposures to an immunogenic stimulus is crucial to mount a more protective immune response upon re-exposure. The antigen-specific adaptive immunity in vertebrates is an example of such a memory to previous immunogenic stimulation. Recently, adaptive characteristics were also attributed to innate immunity, which was classically seen to lack memory. However, the mechanistic details of an adaptive innate immune response are yet to be fully understood and chromatin-based epigenetic mechanisms seem to play an important role in this phenomenon. Possibly, PcG proteins can contribute to such an epigenetic innate immune memory. In this study, we analyzed whether the PcG system can mediate a transcriptional memory of exposure to lipopolysaccharides (LPS). To this end, various forms of LPS pre-treatment were applied to reporter cells and expression kinetics of PcG target genes were analyzed after a second LPS exposure. Neither single nor multiple LPS pre-treatment affected the induction of endogenous LPS-responsive transcripts upon re-exposure. Altogether, our extensive analyses did not provide any evidence for a PcG system-mediated memory of LPS stimulation.

## Introduction

Organisms are regularly exposed to various environmental stimuli including temperature changes, immunogenic substances and other stressors. These environmental factors trigger distinct cellular responses and can have long-lasting impacts on development, metabolism and health, especially when experienced in early life. Exposure to environmental stressors early during development have been shown to affect adult susceptibility to late-onset diseases such as diabetes and cancer [1].

Epidemiological studies have suggested that major effects of environmental exposures on biology could not merely be attributed to genetic characteristics. Rather mechanisms beyond changes in DNA sequence, hence epigenetic changes, contribute to the alteration of

SNF Sinergia grant (CRSII3_160766) (www.snf.
ch). MG received a fellowship from the German
Academic Scholarship Foundation (www.
studienstiftung.de). The funders had no role in
study design, data collection and analysis, decision
to publish, or preparation of the manuscript.

**Competing interests:** The authors have declared
that no competing interests exist.

transcriptional profiles underlying the response and adaptation to environmental conditions [2]. Heritable epigenetic changes, particularly DNA methylation and histone modifications, have been proposed to form the basis of memorizing gene expression states imposed by environmental perturbations.

The Polycomb group (PcG) and Trithorax group (TrxG) proteins form the basis of an epigenetic memory system that remembers and thus maintains developmental signals to keep cellular identities [3]. These proteins bind cis-regulatory DNA elements and act on chromatin via associated chromatin modifiers and components of the transcription machinery. Whereas PcG proteins maintain the transcriptionally silenced state of target genes, TrxG proteins are responsible for the maintenance of active gene expression. PcG proteins react to a variety of environmental stressors [4]. For instance, they are involved in the regulation of responses to heat-shock induced stress and thus could link environmental perturbations and epigenetic memory [5].

In the context of host protection against pathogens, memorizing previous exposures to immunogenic stimuli is of great benefit in order to enable a more protective immune response upon re-exposure, especially in invertebrates like *Drosophila* that lack the T and B lymphocytes-based adaptive immunity. Within the innate immune system, adaptive characteristics are missing by definition. However, this view has been challenged by a growing body of evidence suggesting that pre-exposure to a pathogen can also be memorized by an altered innate immune response to a second exposure [6, 7]. This phenomenon of mounting a more protective innate immune response to a previously encountered pathogen has been termed trained immunity or innate immune memory [8]. Innate immune memory has been described not only in plants [9] and invertebrates that lack the adaptive immune system [10, 11], but also in mammals that do not have functional T and B lymphocytes [12–14]. *Drosophila* pre-exposed to heat-killed bacteria was protected specifically from reinfection with an otherwise lethal dose of the same pathogen throughout their life. This effect was associated with enhanced phagocytosis specifically recognizing and eliminating the previously encountered microbe [10].

Innate immune memory in invertebrates has been associated with different mechanisms like the sustained induction of pathways regulating immune responses, the maintenance of increased levels of positive immune regulators and the altered composition of immune cell populations [6, 15]. Evidence from mammalian studies shows that epigenetic mechanisms can contribute to the memory within the innate immune system, too [16]. In myeloid cells, a repressive chromatin environment usually characterizes inducible immune gene loci in the non-induced state [17, 18]. Upon induction, transcription factors are bound to enhancers and promoters in these regions. This entails the recruitment of cofactors and the subsequent modification of the local chromatin configuration and thus results in regions that are more accessible. This increased chromatin accessibility can be maintained after an initial stimulus and leads to a more efficient response upon a second exposure [19]. The modification of latent enhancers by H3K4 monomethylation upon a first stimulation has been described as another chromatin state that can be maintained and thereby contributes to an altered response upon re-stimulation [20].

As epigenetic mechanisms are instrumental in innate immune memory and PcG proteins react to a variety of external stimuli, it is tempting to hypothesize that PcG proteins might be also involved in epigenetically memorizing previous exposures to immunogenic stimuli. However, up to date, it has not been investigated whether the PcG/TrxG system can mediate a transcriptional memory of exposure to an immunogenic agent in a similar manner as it does in the maintenance of developmental signals. To tackle this question, we analyzed the expression of PcG target genes upon immunogenic stimulation in cells that have been subjected to various forms of pre-stimulation.

For this purpose, *Drosophila* Schneider 2 (S2) cells were an attractive experimental system, as the function of the PcG/TrxG system is highly conserved from flies to humans [3]. S2 cells, that resemble embryonic haemocytes, are a powerful tool to analyze innate immune signaling and regulation in invertebrates [21, 22]. Exposure of S2 cells to lipopolysaccharides (LPS), the principal cell wall components of gram-negative bacteria, has been widely used as a cell culture model to study immune responses to microbial challenges. LPS stimulation causes a distinct transcriptional response in S2 cells [23]. One group of immediate early LPS-inducible genes characterized by peak expression after 1 h includes proapoptotic factors, cytoskeletal and cell adhesion regulators such as Matrix metalloproteinase 1 (Mmp1) and signaling factors like puckered (puc). The second group of early LPS target genes comprises defense and immunity genes including the antibacterial peptides Cecropin A1 (CecA1), Attacin A (AttA) and Metchnikowin (Mtk). Maximal induction levels of these genes are reached after 2 h of LPS stimulation.

In this report, we identified LPS-inducible candidate genes that are targeted by the PcG/TrxG system and whose induction might thus be memorized. Subsequently, the expression of these genes upon LPS stimulation was analyzed by RT-qPCR in cells that have been subjected to various forms of pre-stimulation. In our attempts to identify a PcG system-mediated memory of LPS stimulation, however, we did not find altered responses to re-exposure and thus no indications of a potential memory of previous encounters with LPS.

## Materials and methods

### Identification of LPS-inducible PcG target genes

Genomic positions of PcG binding sites in S2 cells defined by the simultaneous binding of the PcG proteins Polycomb (Pc), Polyhomeotic (Ph) and Posterior sex combs (Psc) were taken from Enderle, D. *et al.* (2011) [24]. Using the R package GenomicRanges and the subsetByOverlaps function [25], the PcG binding sites were overlapped with 500 bp windows around all *Drosophila* transcription start sites (TSS ± 250 nt) which were retrieved from Ensembl using the R package biomaRt [26]. The list of genes for which associated TSS (± 250 nt) overlapped with PcG binding sites was compared to the set of genes differentially expressed upon LPS exposure from Boutros, M. *et al.* (2002) [23]. Genes present in both groups were defined as LPS-inducible PcG target genes.

### Cells and cell culture

*Drosophila* S2-DRSC cells (*Drosophila* Genomics Resource Center, stock #181) were cultured at 25˚C in 100 mm or 145 mm plates (Cellstar Cell Culture Dishes, Greiner Bio-One, Austria) in Schneider's Insect Medium (S0146, Sigma-Aldrich, Switzerland) supplemented with 10% (v/v) fetal bovine serum (FBS, Pansera ES, PAN Biotech, Germany).

### Generation of a stable reporter cell line

For the generation of the reporter cell line, a plasmid containing a GFP reporter whose expression is under the control of the Mmp1 promoter was constructed (see Supplementary Materials and Methods). For transfection, S2 cells were seeded in 6-well plates at a density of 0.9 million cells per ml medium. 400 ng reporter plasmid, which had been linearized by SapI digest, were transfected into cells using the Effectene Transfection Reagent (Qiagen, Germany) according to the manufacturer's instructions. Medium was exchanged the following day and stably transfected cells were selected in the presence of 2 μg/mL Hygromycin B (#10687010, Thermo Fisher Scientific, USA) starting on day 2 post-transfection. Hygromycin-containing

medium was exchanged on alternate days. After having grown to confluence, cells were transferred from 6-well plates to 10 cm plates. When the emergence of drug-resistant colonies was observed, cells were harvested for clonal expansion. By stochastic seeding, single cells were deposited into the wells of a Terasaki plate, filled with conditioned medium. The presence of single cells was verified by microscopy. The cells were gradually cultured in the Terasaki plates in the presence of conditioned medium until cell density allowed for maintenance of the culture under standard conditions.

## LPS treatment

Lipopolysaccharides from *Escherichia coli* 0111:B4 (L4391, Sigma-Aldrich, Switzerland) were dissolved in PBS at a concentration of 1 mg/mL. Aliquots were stored at -20˚C. All LPS treatments were performed at a concentration of 10 μg/mL, except for the low dose follow-up memory experiments. Here, final concentrations of either 1 μg/mL or 0.1 μg/mL LPS were used.

One day prior to LPS stimulation, $2x10^6$ S2 cells in a total of 2 mL medium per well were seeded into 6-well plates. In parallel to LPS treatment, control cells were always exposed to an equal volume of PBS only. For single and multiple LPS pre-treatments, medium was aspirated 2 h after LPS addition, cells were washed twice with 2 mL PBS per well and 2 mL fresh medium was added to each well. In case of multiple LPS pre-treatment, cultures were passaged between the fourth and fifth treatment. The secondary exposure to be analyzed by RT-qPCR was usually carried out three days after the last pre-treatment. Cells were standardly harvested after 30 min, 1 h and 2 h of treatment or first washed with PBS after 2 h and then kept in culture for another 1 h or 2 h.

## Fluorescence microscopy of reporter cells

GFP expression of reporter cell lines was analyzed by fluorescence microscopy on a CKX41 inverted microscope equipped with a U-HGLGPS illumination system (Olympus Life Science, Switzerland).

## FACS analysis of S2 cells

For FACS analysis, WT and reporter cells were cultures in 6-well plates and treated with LPS or PBS as negative control. After different exposure times, cells were harvested and 1 ml cell suspension was used for FACS analysis on a BD LSR Fortessa FACS analyzer. Filters used were 530/30 GFP 488 (C)-A and 610/20 mCherry 561(D)-A. All FACS analyses were performed on three biological replicates.

## RNA extraction from S2 cells and cDNA synthesis

For RT-qPCR experiments, total RNA was isolated from cells cultured in 6-well plates. To extract RNA, cells were harvested in 1 mL TRIzol Reagent (Ambion, USA) and transferred into 1.5 mL reaction tubes. Samples were stored at -80˚C until further processed. After thawing, samples were centrifuged at 21130 rcf for 10 min at 4 ˚C and RNA was extracted from 700 μL of the supernatant using the Direct-zol RNA MiniPrep kit (Zymo Research, USA) according to the manufacturer's instructions. To remove residual genomic DNA, DNase I in-column treatment was performed as recommended by the manufacturer. Differing from the suggested protocol, RNA was eluted from the column in 36 μl nuclease-free water. Purity and integrity of isolated RNA was assessed on a 2100 Bioanalyzer System (Agilent Technologies,

USA) or a spectrophotometer (NanoPhotometer NP80, Implen, Germany). The latter was also used for quantification of RNA.

cDNA was synthesized from 500 ng RNA in a total reaction volume of 10 μL using the First Strand cDNA Synthesis Kit (Thermo Fisher Scientific, USA) as suggested by the manufacturer with the following adaptations: A 1:1 mixture of random hexamer primers and oligo(dT)18 primers was used and the cDNA synthesis was performed for 5 min at 25 ˚C followed by 60 min at 43 ˚C before terminating the reaction by heating at 70˚C for 5 min. Negative controls without reverse transcriptase (minus RT controls) were processed in parallel.

### Quantitative RT-PCR of cDNA samples

Quantitative RT-PCR of cDNA samples from S2 cells was performed with a SYBR Green I-based reaction mix (FastStart Essential DNA Green Master (Roche Life Science, Switzerland)) and gene-specific primer pairs (designed using Primer3 software and NCBI Primer BLAST) (S1 File of S1 Table) on a LightCycler 96 Instrument (Roche Life Science, Switzerland). Reaction mixtures included 5 μL 2x FastStart Essential DNA Green Master, 1 μL primer mix (5 μM forward primer, 5 μM reverse primer) and 4 μL cDNA diluted 1:50 in nuclease-free water. All reactions were run in triplicates in 96-well plates and prepared using a repeater pipette (Repeater Xstream pipette, Eppendorf, Germany). qPCR conditions included a preincubation step at 95 ˚C for 10 min and 45 cycles of a 3-step-amplification consisting of 95 ˚C for 10 s, 60 ˚C for 10 s and 72 ˚C for 10 s. qPCR data was analyzed using the comparative ΔΔCt method [27]. Expression levels of Ribosomal protein L32 (RpL32) were used for normalization. Robustly constant expression of RpL32 across samples was verified by quantification relative to a second endogenous reference gene, ATPsynCF6.

## Results

### PcG binding is enriched at transcription start sites of certain LPS-inducible genes in S2 cells

LPS-inducible genes that are regulated by the PcG/TrxG system are potential candidate genes whose expression might be affected by previous exposures to LPS and thereby memorized. To identify such candidate genes, two published datasets were used: data from chromatin immunoprecipitation of PcG proteins followed by sequencing (ChIP-Seq) [24] and gene expression microarray data from LPS-treated S2 cells [23]. PcG binding sites defined by the simultaneous binding of the PcG proteins Polycomb (Pc), Polyhomeotic (Ph) and Posterior sex combs (Psc) were computationally overlapped with the transcription start sites (TSS) of LPS-inducible genes. In total, 24 out of the 223 LPS-induced genes (10.8%) identified by Boutros, M. *et al.* (2002) [23] were bound by PcG proteins in a range of ± 250 nucleotides around their TSS. PcG binding was clearly enriched in this subset of genes affected by LPS stimulation (p = 4.825e-08 (hypergeometric test); p = 5.476e-08 (Fisher's Exact Test)) compared to around 2.6% of all *Drosophila* genes that were bound by PcG proteins. This might suggest a potential functional link between PcG-mediated gene regulation and activation of target genes by LPS. Genes differentially expressed upon LPS exposure and bound by PcG proteins around their TSS included cytoskeletal and cell adhesion modulators such as Matrix metalloproteinase 1 (Mmp1) and signaling proteins such as puckered (puc) and unpaired 2 (upd2) (Table 1). Most PcG target genes identified were upregulated by LPS. This is in line with the hypothesis that PcG silencing might be impaired by a first exposure and thus target genes might be induced more strongly upon subsequent exposures.

**Table 1. PcG target genes that were differentially expressed upon LPS exposure.**

| Gene Symbol | Gene Name | Annotation Symbol | Molecular Function |
|---|---|---|---|
| Adgf-A | Adenosine deaminase-related growth factor A | CG5992 | adenosine deaminase activity, growth factor activity |
| chn | charlatan | CG11798 | DNA-binding transcription repressor activity, sequence-specific DNA binding, protein binding |
| CrebB | Cyclic-AMP response element binding protein B | CG6103 | sequence-specific DNA binding |
| Dronc | Death regulator Nedd2-like caspase | CG8091 | cysteine-type endopeptidase activity, protein homodimerization activity |
| edl | ETS-domain lacking | CG15085 | activating transcription factor binding, repressing transcription factor binding |
| IP3K1 | Inositol 1,4,5-triphosphate kinase 1 | CG4026 | inositol-1,4,5-trisphosphate 3-kinase activity |
| Keap1 | Keap1 | CG3962 | actin binding |
| Mmp1 | Matrix metalloproteinase 1 | CG4859 | metalloendopeptidase activity |
| PGRP-LF | Peptidoglycan recognition protein LF | CG4437 | peptidoglycan binding |
| Ptpmeg2 | Protein tyrosine phosphatase Meg2 | CG32697 | non-membrane spanning protein tyrosine phosphatase activity |
| puc | puckered | CG7850 | JUN kinase phosphatase activity |
| pyd | polychaetoid | CG43140 | cell adhesion molecule binding |
| pyr | pyramus | CG13194 | fibroblast growth factor receptor binding |
| QC | Glutaminyl cyclase | CG32412 | glutaminyl-peptide cyclotransferase activity |
| Sema-5c | Semaphorin-5c | CG5661 | semaphorin receptor binding |
| Ser | Serrate | CG6127 | Notch binding |
| Sesn | Sestrin | CG11299 | leucine binding |
| shn | schnurri | CG7734 | DNA-binding transcription factor activity, RNA polymerase II activating transcription factor binding, transcription coactivator activity |
| subdued | subdued | CG16718 | intracellular calcium activated chloride channel activity |
| Svil | Supervillin | CG33232 | actin binding |
| TepII | Thioester-containing protein 2 | CG7052 | endopeptidase inhibitor activity |
| tna | tonalli | CG7958 | zinc ion binding |
| upd2 | unpaired 2 | CG5988 | cytokine activity |
| yin | yin | CG44402 | proton-dependent oligopeptide secondary active transmembrane transporter activity |

## Mmp1 transcripts are only induced in a subset of S2 cells upon LPS exposure

A large fraction of cells responding to LPS treatment is the basic prerequisite in order to be able to detect a potential memory of LPS exposure in bulk cell culture in terms of an altered response to re-exposure. Otherwise, only a small fraction of cells will have reacted to both a first and a second exposure and only these ones will have the potential to show a memory effect. However, this potential effect might get lost in the majority of unresponsive cells. Therefore, it is crucial for a biochemically assessed memory experiment to know the approximate proportion of responding cells. For this purpose, fluorescent *in situ* hybridization (FISH) experiments for Mmp1 transcripts were performed to first analyze how many S2 cells do actually respond to LPS stimulation. Control cells treated with phosphate-buffered saline (PBS) contained only very few FISH spots indicative of single Mmp1 transcripts. However, after 1 h LPS treatment, the number of Mmp1 transcripts was strongly increased, but only in approximately 25% of exposed single cells (S1 File of S1A, B Fig). Most cells did not show an elevated number of fluorescent spots after LPS treatment in FISH experiments and rather looked like PBS-treated control cells. In contrast, a large number of Act5C mRNA molecules was detected in every cell (S1 File of S1C Fig). This meant that the FISH method was principally capable of

labelling transcripts in every single cell. Therefore, technical limitations of the *in situ* hybridization method could be ruled out as a reason for Mmp1 transcripts being detected in a subset of S2 cells only. Consequently, these results indicate that only a minor fraction of cells induces LPS-responsive gene expression upon LPS stimulation.

## Generation of a reporter cell line from single clones that express GFP under the control of the Mmp1 promoter

The *in situ* hybridization experiments showed that only a minor fraction of cells induces LPS-responsive gene expression upon LPS stimulation. To be able to select S2 cells that have reacted to LPS and thus get an accurate picture of cell responses to repeated LPS treatment, reporter cell lines that induce green fluorescent protein (GFP) expression upon LPS treatment were generated. Thereby, it will be possible to sort S2 cells that will have reacted to LPS and thus express GFP. The isolated fraction of responders can be subjected to a second LPS stimulus to see whether pre-treatment affects the response to a second exposure and therefore is memorized. This will be a targeted analysis as only the fraction of cells, that actually has reacted to LPS the first time and therefore has the potential to memorize this first exposure, will be exposed a second time.

To generate such a reporter cell line, a plasmid containing a GFP reporter whose expression is under the control of the Mmp1 promoter was constructed (S1 File of S2 Fig). The Mmp1 promoter region was chosen to control GFP expression because it was found to be the top inducible LPS target gene of the immediate early group in the transcriptomic analysis [23] (see also S1 File of S2 Table) and identified as a PcG target (Table 1). The 4.78 kb Mmp1 promoter region to be fused to the GFP gene has been described before to comprise three AP-1 binding sites and to functionally regulate Mmp1 expression. It has been previously used to generate transgenic flies regulating the expression of a *lacZ* reporter gene. Reporter expression in these lines perfectly matched the expression pattern of endogenous Mmp1 and the reporter was upregulated in wing discs upon activation of c-Jun N-terminal kinase (JNK) [28].

The GFP reporter plasmid was used to generate monoclonal Mmp1 reporter cells. Several clones were obtained and tested by fluorescence microscopy for GFP inducibility upon LPS exposure. LPS treatment led to GFP fluorescence in the majority of cells in several monoclonal Mmp1 reporter cell lines (Fig 1A).

## Mmp1 promoter-controlled GFP expression is induced in majority of reporter cells upon LPS treatment

In one Mmp1 reporter line, the response to LPS was characterized to a greater extent. Reporter cells were exposed to LPS, harvested after different time points and subjected to fluorescence-activated cell sorting (FACS) to analyze GFP reporter expression. LPS treatment induced GFP fluorescence in the majority of reporter cells, while it was absent in wild-type (WT) control cells. 84% of the reporter cells showed GFP fluorescence already after 2 h of LPS treatment and 90% of the cells were GFP-positive 6 h after LPS addition (Fig 1B and 1C). Not only the frequency of GFP-expressing reporter cells, but also their GFP intensity increased with exposure time. Fluorescence intensity of GFP-positive cells was higher after 4 h LPS exposure as compared to 2 h treatment and LPS exposure for 6 h did not further increase GFP intensity of reporter cells (Fig 1D).

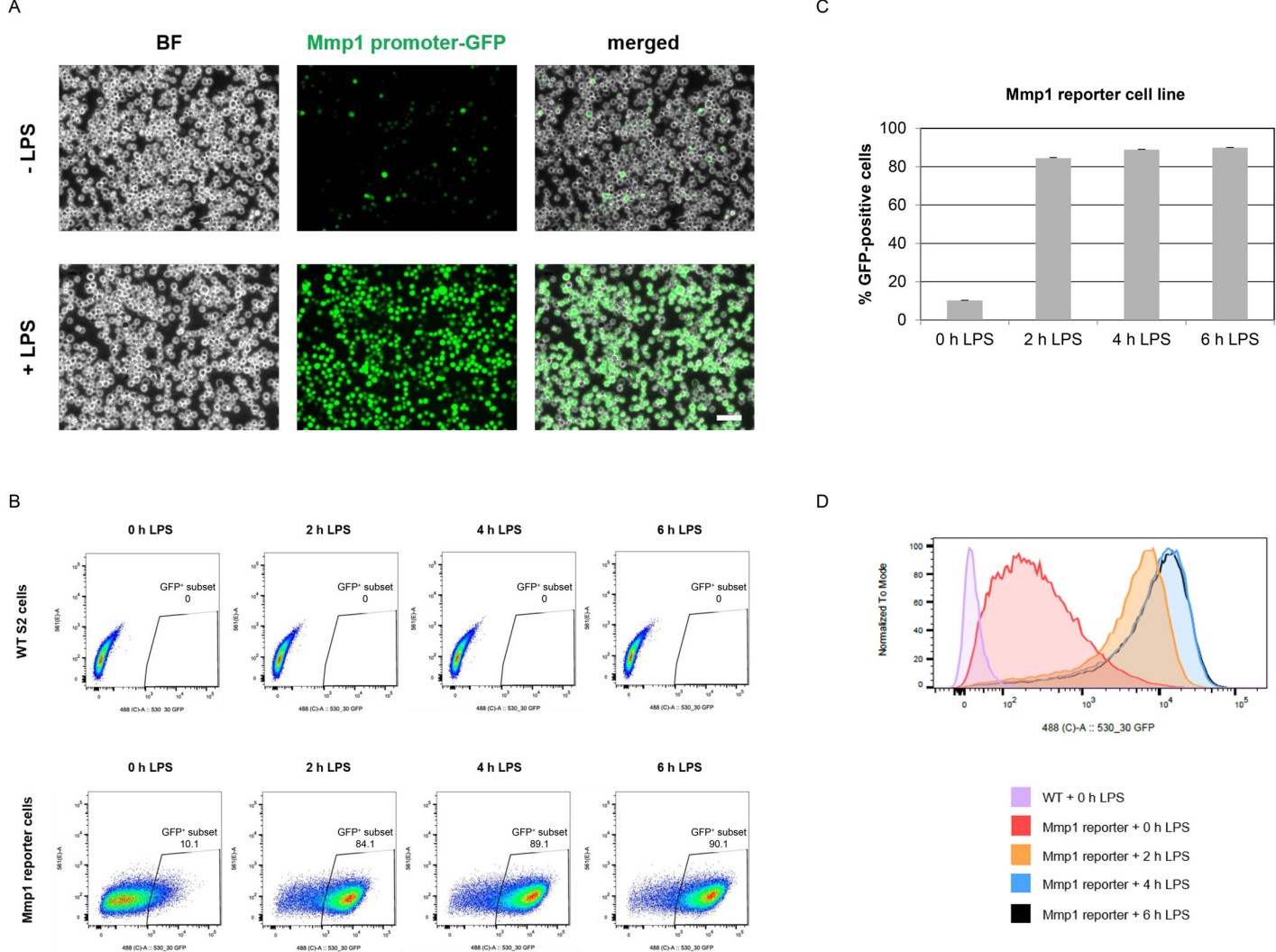

**Fig 1. Analysis of GFP fluorescence in Mmp1 reporter cells exposed to LPS by fluorescence microscopy and FACS.** (A) Monoclonal Mmp1 reporter cells with GFP under the control of the Mmp1 promoter were exposed to LPS for 6 h and visualized by fluorescence microscopy. Control cells were treated with PBS only. Scale bar: 50 μm. (B, C, D) GFP fluorescence in WT and Mmp1 reporter cells treated with LPS for 0 h, 2 h, 4 h and 6 h was assessed by FACS. (B) Dot plots of GFP fluorescence in WT and Mmp1 reporter cells and (C) frequency of GFP-positive cells in the reporter line are shown. Mean percentage from three biological replicates are given. Error bars represent standard deviations. (D) The frequency distribution of the FACS data versus the GFP intensity is displayed in a histogram.

## GFP mRNA and transcripts of endogenous LPS target genes are strongly induced in the reporter cell line upon LPS treatment

To look at GFP induction not only at protein but also at transcript level, GFP mRNA expression was analyzed in reporter cells exposed to LPS by RT-qPCR. GFP transcripts were rapidly induced after LPS treatment and reached peak expression with 51-fold and 56-fold induction after 1 h and 2 h, respectively, with respect to non-treated reporter cells (Fig 2A). Afterwards, GFP mRNA levels dropped again markedly.

Next, the expression of endogenous Mmp1 transcripts and further LPS target genes was evaluated in both reporter and WT S2 cells. LPS exposure rapidly induced Mmp1 mRNA expression in reporter cells (Fig 2B). Expression levels were maximal after 1 h of treatment with a 30-fold induction compared to untreated WT cells. The induction in reporter cells

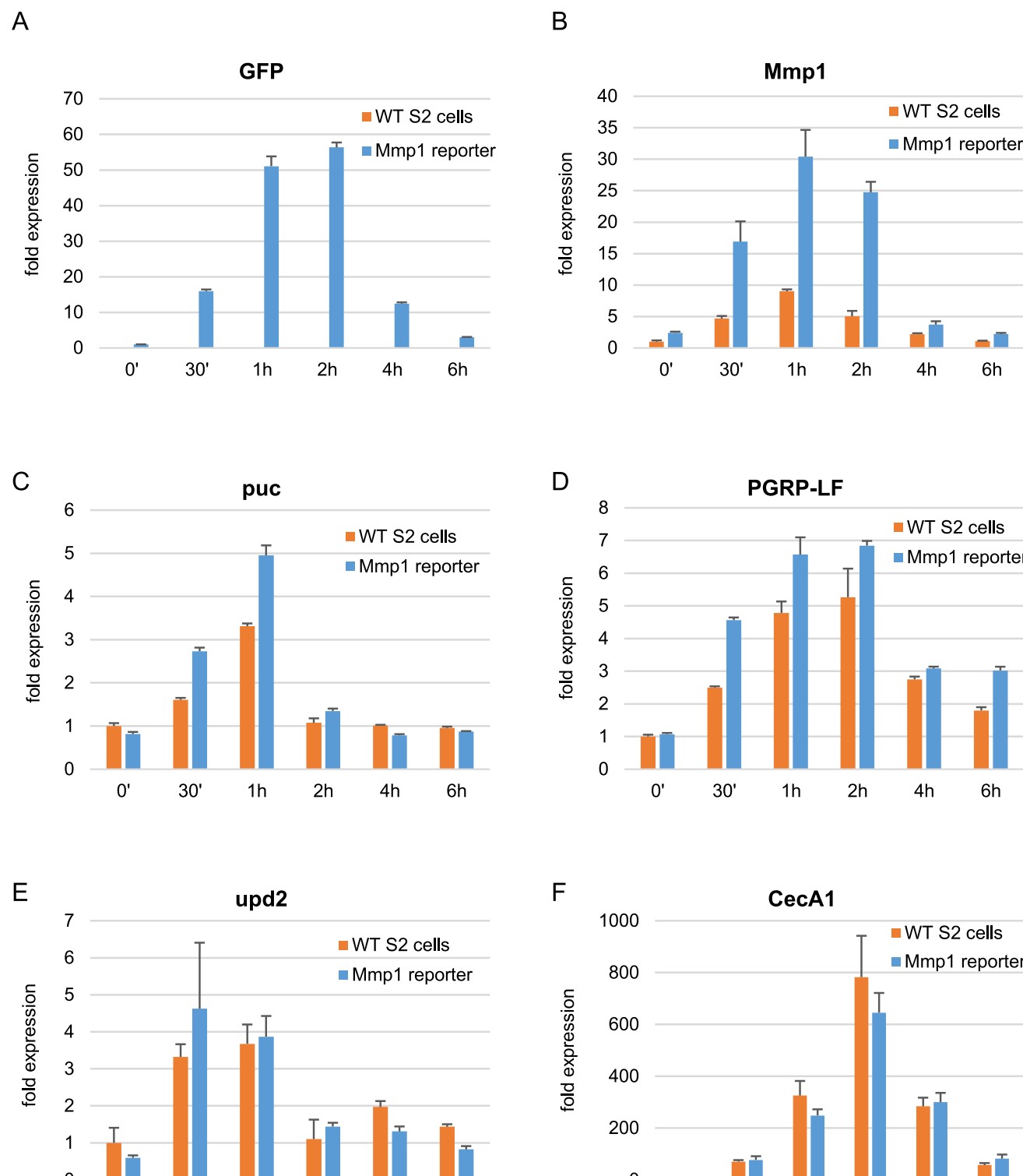

**Fig 2. Expression of LPS-inducible genes in WT and Mmp1 reporter cells upon LPS treatment for 0 min, 30 min, 1 h, 2 h, 4 h and 6 h.** mRNA expression of (A) GFP, (B) Mmp1, (C) puc, (D) PGRP-LF, (E) upd2 and (F) CecA1 was analyzed by RT-qPCR. Expression levels were normalized to Ribosomal protein L32 (RpL32) levels and the 0 min time point values in WT S2 cells. As GFP mRNA was absent in WT S2 cells, GFP levels were normalized to the 0 min time point in Mmp1 reporter cells instead. Robustly constant expression of RpL32 across samples was verified by quantification relative to a second endogenous reference gene, ATPsynCF6 (compare S1 File of S5A Fig). Mean fold changes from three biological replicates are shown. Error bars represent standard deviations.

followed the same kinetics as in WT cells. Peak expression was reached after 1 h of treatment and subsequently transcript levels decreased until coming back to uninduced levels after only 6 h. Interestingly, both uninduced and induced Mmp1 levels were higher in reporter cells as compared to WT S2 cells. In the untreated case, Mmp1 levels were 2.4-fold increased in reporter cells as compared to WT S2 cells. At the time points of highest induction, Mmp1 expression in reporter cells was even 3.4-fold (1 h) and 4.9-fold (2 h), respectively, higher than in WT S2 cells.

On top of Mmp1 levels, expression of three additional LPS-inducible PcG target genes (puc, PGRP-LF, upd2) and of a non-target gene (CecA1) were evaluated by RT-qPCR. puc and PGRP-LF transcripts were stronger increased in reporter cells as compared to WT S2 cells, albeit sharing similar induction kinetics (Fig 2C and 2D). In comparison, upd2 was more similarly induced in both cell lines (Fig 2E). Intriguingly, basal levels of puc, PGRP-LF and upd2 were not elevated in reporter cells as compared to WT S2 cells. This was in contrast to Mmp1, whose expression was 2.4-fold stronger in uninduced reporter cells than in WT S2 cells. Furthermore, CecA1 was induced in a like manner by LPS in both cell lines (Fig 2F).

These FACS and RT-qPCR experiments showed that nearly all cells induce GFP protein expression upon LPS stimulation and that endogenous LPS target genes were induced with similar kinetics, but to different maximum levels in reporter cells. Therefore, there was no need to sort the cells by GFP expression and isolate the fraction of responders to a first stimulation for subsequent re-stimulation in memory experiments. Mmp1 reporter cell lines can directly be used to accurately analyze the cellular response to repeated LPS exposure, as the fraction of responders will be very high.

To make sure that the candidate genes, whose expression might be affected by previous exposures to LPS and thereby memorized, are also targeted by the PcG system in the Mmp1 reporter cell line, Pc binding to candidate gene loci was verified by chromatin immunoprecipitation (ChIP) experiments. The Pc protein was bound to the Mmp1 locus, for example, to a similar extent in the Mmp1 reporter cell line as in WT S2 cells (S1 File of S3 Fig).

## Single LPS pre-treatment does not affect induction of GFP and endogenous LPS-responsive genes 72 h later in both reporter and wild-type cells

To test for a potential memory of pre-exposure to LPS, 2 h of pre-treatment were applied to the fluorescent reporter cells and expression kinetics of LPS-inducible genes were analyzed after a second exposure. A memory of the initial exposure will be maintained, if re-stimulation induces an altered and adapted second response. Alterations can affect different aspects of the response. Target genes can be more strongly induced reaching higher expression levels upon re-exposure. Previous stimulation can influence the induction or termination kinetics of a later response when cells re-encounter the immunogenic agent or the duration of stimulus-induced gene expression is prolonged upon re-exposure.

In these experiments, LPS was administered to reporter cells and washed off after 2 h. The pre-exposed cells were further cultured in new pre-conditioned medium for 72 h. During this time, S2 cells usually divide 3–4 times and hence a potential memory of LPS exposure would need to be maintained during cell division in order to be detected. Cells were then re-stimulated with LPS and gene expression was analyzed by RT-qPCR at different time points after re-stimulation (Fig 3A). Evaluated time points included 30 min, 1 h and 2 h of LPS re-stimulation as well as 1 h and 2 h after washing off LPS from a 2 h re-exposure.

In these experiments, Mmp1 promoter-mediated GFP induction was the very same in pre-treated reporter cells as compared to cells that have never been exposed to LPS before (Fig 3B). 2 h LPS pre-treatment neither affected the expression of endogenous Mmp1, when stimulated again, in both reporter and WT S2 cells (Fig 3C). Similarly, the LPS-inducible PcG target genes

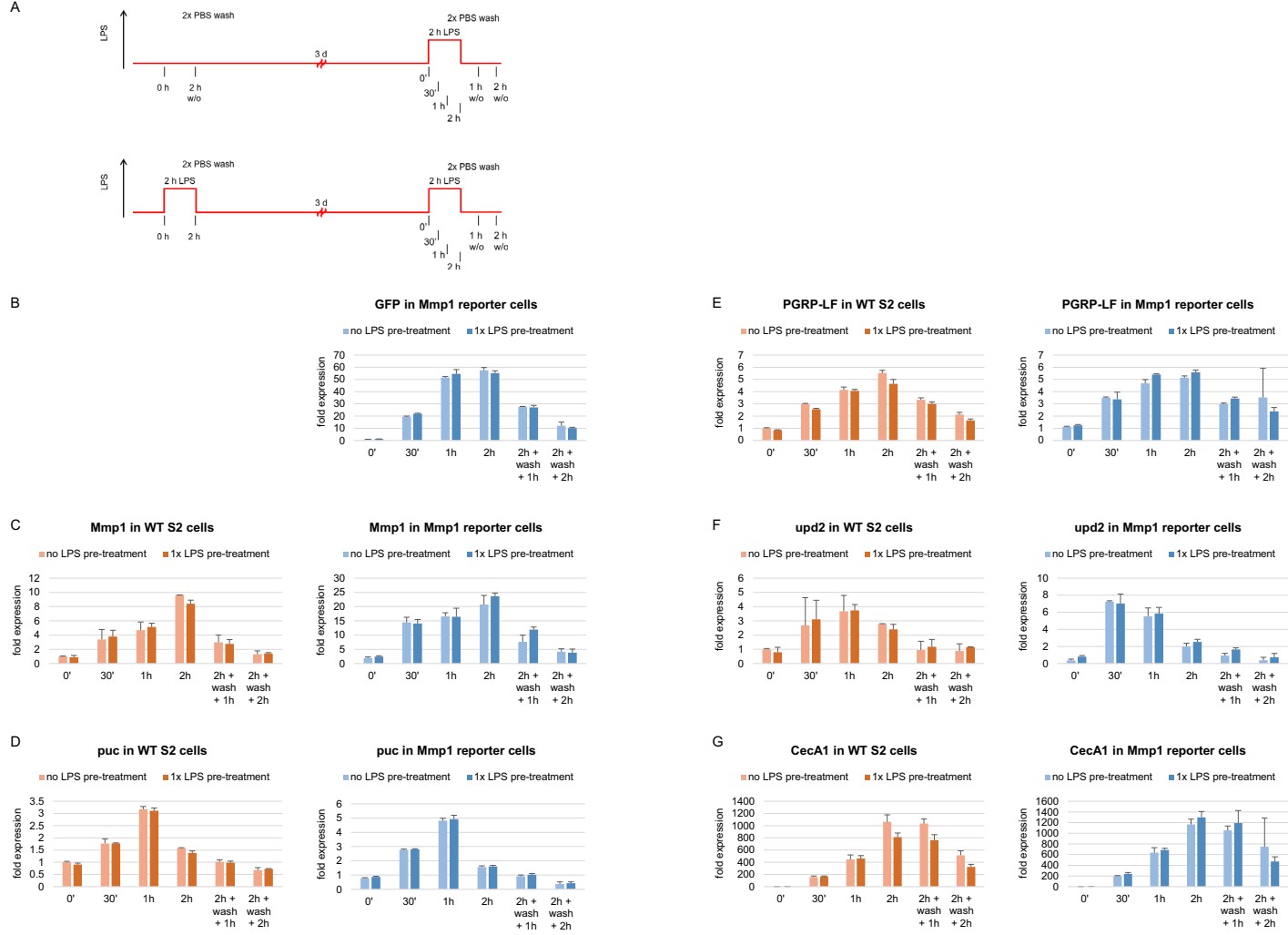

**Fig 3. Effects of single LPS pre-treatment on the expression of LPS-inducible genes upon another LPS treatment three days later in WT and Mmp1 reporter cells.**
(A) Experimental outline of the single pre-treatment. Cells were stimulated for 2 h, cultured for another three days, re-stimulated with LPS and subjected to RT-qPCR analysis after different time points. Transcript levels of (B) GFP, (C) Mmp1, (D) puc, (E) PGRP-LF, (F) upd2 and (G) CecA1 were evaluated after the second LPS exposure in both WT and Mmp1 reporter cells either pre-treated or not pre-exposed to LPS before. Expression levels were normalized to Ribosomal protein L32 (RpL32) levels and the 0 min time point values in WT S2 cells that were not pre-treated. As GFP mRNA was absent in WT S2 cells, GFP levels were normalized to the 0 min time point in non-pre-treated Mmp1 reporter cells instead. Robustly constant expression of RpL32 across samples was verified by quantification relative to a second endogenous reference gene, ATPsynCF6 (compare S1 File of S5B Fig). Mean fold changes from three biological replicates are shown. Error bars represent standard deviations.

puc, PGRP-LF and upd2 and the control gene CecA1 showed each comparable induction kinetics, independent of pre-exposure to LPS, in both cell lines (Fig 3D, 3E, 3F and 3G).

These experiments demonstrated that a single LPS pre-treatment for 2 h does not affect the induction of LPS-responsive genes 72 h later in both reporter and WT cells.

## Multiple LPS pre-treatment does not affect induction of GFP and endogenous LPS-responsive genes 72 h later in both reporter and wild-type cells

Hypothesizing that a one-time pre-treatment might be a trigger too small to be remembered, the pre-treatment was extended to 2 h-periods of LPS exposure each on five consecutive days.

For these experiments, cells were treated with LPS and washed twice with PBS to remove LPS again after 2 h. This treatment was redone 22 h later, i. e. 24 h after starting the previous one, in total five times on five consecutive days. Pre-exposed cells were further cultured, then re-stimulated with LPS for different time periods and subjected to FACS analysis to evaluate GFP fluorescence (S1 File of S4A Fig) and RT-qPCR evaluation to determine induction kinetics of LPS-inducible transcripts upon re-stimulation (Fig 4A).

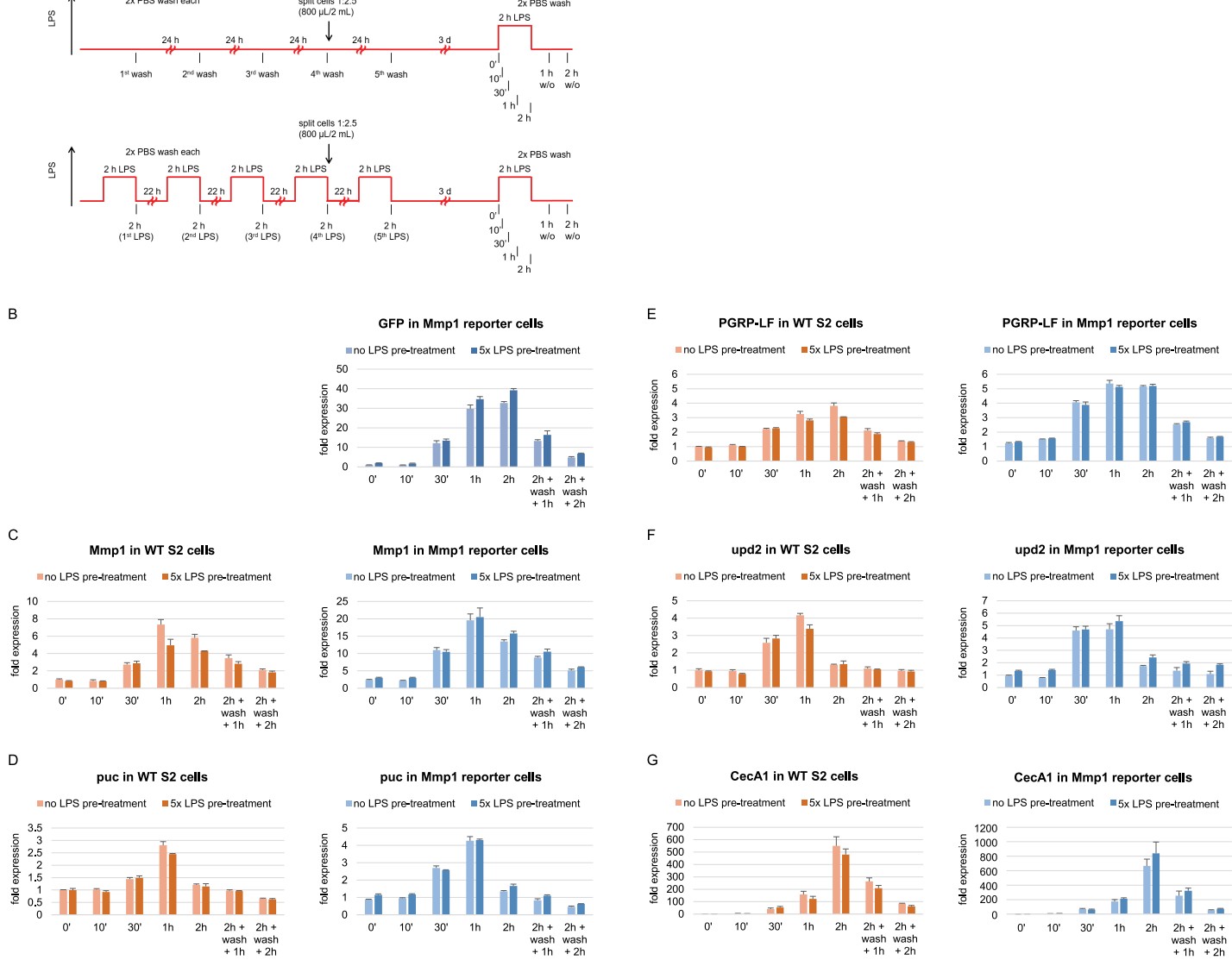

**Fig 4. Effects of multiple LPS pre-treatment on the expression of LPS-inducible genes upon another LPS treatment three days later in WT and Mmp1 reporter cells.** (A) Experimental outline of the multiple pre-treatment. Cells were stimulated for 2 h each on five consecutive days, cultured for another three days, re-stimulated with LPS and subjected to RT-qPCR analysis after different time points. Transcript levels of (B) GFP, (C) Mmp1, (D) puc, (E) PGRP-LF, (F) upd2 and (G) CecA1 were evaluated after the second LPS exposure in both WT and Mmp1 reporter cells either pre-treated or not pre-exposed to LPS before. Expression levels were normalized to RpL32 levels and the 0 min time point values in WT S2 cells that were not pre-treated. As GFP mRNA was absent in WT S2 cells, GFP levels were normalized to the 0 min time point in non-pre-treated Mmp1 reporter cells instead. Robustly constant expression of RpL32 across samples was verified by quantification relative to a second endogenous reference gene, ATPsynCF6 (compare S1 File of S5C Fig). Mean fold changes from three biological replicates are shown. Error bars represent standard deviations.

Before starting the re-exposure, more cells that still showed GFP fluorescence were detected in the cultures that had been pre-treated as compared to untreated control cells. Not all pre-treated cells had completely lost GFP fluorescence within the subsequent culturing period after the last pre-treatment. Despite this difference in the number of GFP-positive cells when starting the re-exposure, no major differences were observed in GFP induction between reporter cells that have been pre-treated and the ones that have never been exposed before (S1 File of S4B, C Fig). Similarly, GFP intensity of reporter cells was not altered between the two conditions (S1 File of S4D, E Fig). Overall, this FACS analysis revealed that multiple LPS pre-treatment has no effect on GFP fluorescence upon re-exposure, except for different starting levels derived from residual fluorescence of the pre-treatment.

Similarly, it was analyzed how multiple pre-treatment affects the transcript levels of LPS-inducible genes after a second exposure (Fig 4A). Not only GFP fluorescence, but also GFP mRNA levels were slightly elevated in reporter cells before the start of the re-treatment (Fig 4B). Three days after washing off LPS from the last pre-treatment, starting GFP levels were 2-fold higher in pre-treated cells than in control cells. However, GFP mRNA expression reached the same level again after only 30 min of re-exposure in both non-pre-treated and pre-treated cells. At later time points of LPS re-exposure as well as after washing off LPS, GFP mRNA levels were marginally elevated in pre-treated cells. However, the difference was much too small to be considered as a memory effect of LPS pre-treatment.

The induction of endogenous Mmp1 expression, when stimulated again, was neither affected by multiple pre-treatment in both reporter and WT S2 cells (Fig 4C). Its induction kinetics were independent of the pre-treatment. Similar results were obtained for the other LPS-inducible genes tested. The induction of the PcG target genes puc, PGRP-LF and upd2 and the control gene CecA1 upon re-stimulation was not affected by pre-exposure to LPS in both cell lines either (Fig 4D, 4E, 4F and 4G).

All in all, these experiments demonstrated that a multiple LPS pre-treatment does not affect the induction of neither GFP nor endogenous LPS-responsive genes triggered by a second stimulus 72 h after the last pre-treatment pulse.

## Multiple LPS pre-treatment does not affect low dose induction of GFP and endogenous LPS-responsive genes 72 h later in both reporter and wild-type cells

In the previously described experiments, a final LPS concentration of 10 μg/mL LPS was used for all LPS stimulations. This is a concentration commonly used in S2 cells which leads to a strong induction of LPS-responsive genes. This might be such a strong stimulus that cells always respond with maximal strength, independently of a previous LPS exposure history, in order to be able to successfully fight an infection which is mimicked by LPS treatment. Yet it might be possible that cells get sensitized by LPS pre-treatment and pre-exposed cells show a stronger second response to low LPS doses which would normally induce LPS-responsive genes either only minimally or not at all. To test this hypothesis, 10 μg/mL LPS were applied to reporter cells for 2 h each on five consecutive days. A second response induced by small LPS concentrations of 1 μg/mL or 0.1 μg/mL LPS was subsequently analyzed. The first one only leads to a much-attenuated response, while the latter—representing only a hundredth of the usual LPS amount—normally does not induce most LPS-responsive genes at all.

Indeed, the control gene CecA1 which reaches a maximum induction of around 666-fold after 2 h when stimulated with 10 μg/mL LPS, was only induced 83-fold and 17-fold, respectively, when exposed to 1 μg/mL or 0.1 μg/mL LPS (Fig 5F). GFP mRNA levels were elevated to 3-fold increase after 2 h exposure to 1 μg/mL and even not induced by 0.1 μg/mL LPS

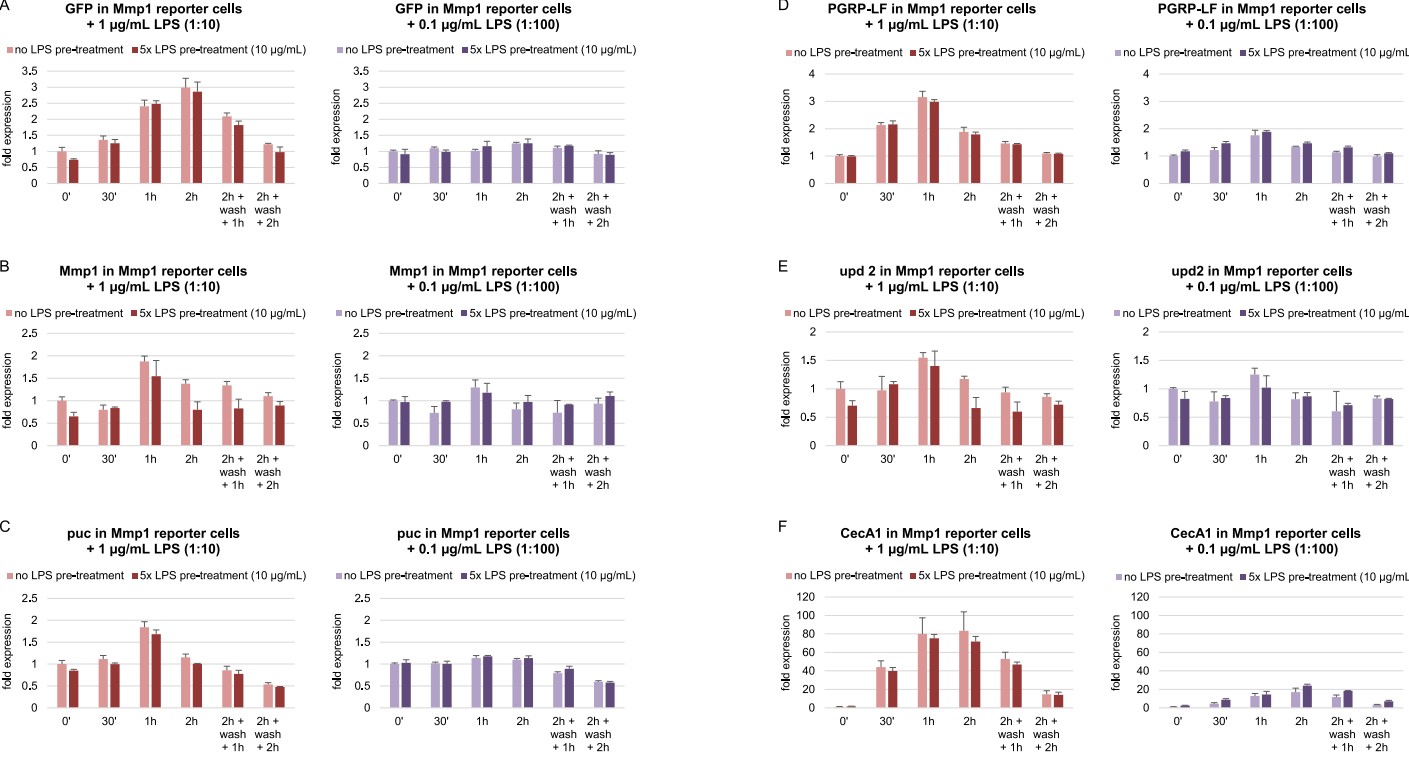

**Fig 5. Effects of multiple LPS pre-treatment on the expression of LPS-inducible genes upon another low-dose LPS treatment three days later in Mmp1 reporter cells.** Mmp1 reporter cells were exposed to LPS for 2 h each on five consecutive days, cultured for another three days and re-stimulated with either 1 μg/mL or 0.1 μg/mL LPS for different time points. Transcript levels of (A) GFP, (B) Mmp1, (C) puc, (D) PGRP-LF, (E) upd2 and (F) CecA1 were evaluated by RT-qPCR after the low-dose LPS exposures in cells either pre-treated or not pre-exposed to LPS before. Expression levels were normalized to RpL32 and the 0 min time point values in cells that were not pre-treated. Robustly constant expression of RpL32 across samples was verified by quantification relative to ATPsynCF6 (compare S1 File of S5D Fig). Mean fold changes from three biological replicates are shown. Error bars represent standard deviations.

(Fig 5A). By way of comparison, GFP was induced 33-fold upon 2 h stimulation with 10 μg/mL LPS. However, the effect of the low dose LPS treatment on GFP transcripts was independent of multiple pre-treatment. Low dose induction kinetics of GFP mRNA were the very same in reporter cells that were exposed to five times pre-treatment with 10 μg/mL LPS as compared to non-pre-treated control cells. Endogenous Mmp1 expression was barely induced by low LPS concentrations and pre-exposed cells did not show a stronger second response to low LPS doses (Fig 5B). puc, PGRP-LF and upd2 responses to reduced LPS concentrations were also unaffected in reporter cells exposed to multiple LPS pre-treatment as compared to cells that were never pre-treated before (Fig 5C, 5D and 5E).

These findings indicate that LPS treatment of S2 cells is neither memorized in terms of sensitization which would result in a strong induction of LPS-responsive genes even when a second response occurs with a low LPS dose only.

## Discussion

In an organism like *Drosophila* lacking a system of adaptive immunity, it would be beneficial to memorize a previous exposure to an immunogenic stimulus in order to be able to mount a more protective immune response upon re-encounter. Chromatin-based epigenetic

mechanisms like the PcG/TrxG system might underlie such a potential memory to previous immunogenic stimulation within the innate immune system.

For the analysis of such an immune memory in a *Drosophila* cells system, a monoclonal S2 reporter cell line that expressed GFP under the control of the LPS-inducible Mmp1 promoter was generated in this study. As only a small fraction of S2 WT cells responded to LPS treatment, this cell line allowed to study the response to multiple exposures in a well-controlled homogenous system. More than 90% of the reporter cells induced GFP expression upon LPS stimulation. On top of GFP reporter expression, endogenous transcripts of LPS-responsive genes were also induced in the reporter cell line with similar kinetics as in WT S2 cells. Interestingly, basal as well as induced levels of several endogenous transcripts like Mmp1 were higher in the reporter cell line as compared to WT S2 cells. This was another indication that a much larger fraction of reporter cells had the potential to induce and actually did induce endogenous LPS-responsive genes in comparison to WT S2 cells. The difference in induction levels upon LPS exposure between reporter cells and WT S2 cells was largest for Mmp1. For the other LPS-inducible PcG target genes analyzed in this study, induction levels were not as much elevated in the reporter cell line, which might point to more similar fractions of cells in the two cell lines that induce these genes upon LPS exposure.

Another point worthy of note is the rather low induction level of the analyzed PcG target genes as compared to the around 1000-fold upregulation of the control gene CecA1 upon LPS stimulation. Therefore, it is tempting to speculate that the PcG-dependent chromatin state might dampen the induction level of target genes.

To test for a potential PcG-mediated memory of pre-exposure to LPS, various forms of LPS pre-treatment were applied to newly generated fluorescent reporter cells and expression kinetics of LPS-inducible PcG target genes were analyzed after a second exposure. However, neither single nor multiple LPS pre-exposure affected the induction of GFP and the expression of endogenous LPS-responsive transcripts when stimulated again. Additionally, LPS pre-treatment neither sensitized the reporter cells in such a way that they would show a stronger second response to low LPS doses, which would normally induce LPS-responsive genes either only minimally or not at all. Overall, the presented experiments did not provide any evidence that a response to LPS is memorized in S2 cells in terms of an altered cellular response to a second stimulus.

These findings are based on carefully controlled experiments each performed at least in biological triplicates, considering technical limitations like the induction of LPS-responsive genes in only a fraction of S2 WT cells. However, they do not rule out that there is a memory component to the innate immune system in *Drosophila* mediated by PcG/TrxG proteins-based epigenetic mechanisms. This study focused on LPS-inducible genes directly targeted by the PcG system. It is also possible that pre-exposure to LPS influences the induction of other LPS-inducible genes upon re-stimulation. Their expression might be under the control of a factor, which in turn could be itself regulated by the PcG/TrxG system.

On the other hand, a potential memory of previous exposure to immunogenic stimuli might be formed by epigenetic mechanisms other than the PcG/TrxG system. For example, it has been reported that previously unmethylated distant enhancer regions become mono-methylated at histone residues H3K4 upon LPS exposure in mouse bone marrow-derived macrophages and that this histone mark is maintained after the LPS stimulus [20]. In *Drosophila* haemocytes, an RNA interference (RNAi) mechanism based on small interfering RNAs has also been shown to keep an immune memory against viruses [29].

One can also speculate that a functional transcriptional memory of previous exposures might require an organismal context and is not effective in isolated cell cultures. If so, similar experiments to the ones performed in cell culture in this study can be adapted to the

developing fly. For example, bacteria can be injected in the abdomen of adult flies [10]. In third instar larvae, an infection can be triggered by pricking larvae with a needle previously inoculated with bacteria [30]. Bacterial infection can also be induced naturally by oral infection of larvae and adults with bacteria-containing food [31].

The presented results might also indicate that an epigenetic memory in the part of innate immunity, which involves the LPS-inducible genes analyzed in this study, is not required. The rapid gene induction system in the investigated cell system can be sufficient to maintain immunogenic responses strong enough to efficiently protect a host from an infection and thus make an epigenetic contribution unnecessary.

## Supporting information

**S1 File.**
(PDF)

## Acknowledgments

MG is a member of the Life Science Zurich Graduate School (PhD Program in Molecular Life Sciences) and supported by the German Academic Scholarship Foundation. We thank the D-BSSE Single Cell Facility for assistance with microscopy and FACS analysis and Tosca Birbaumer for help with ChIP experiments.

## Author Contributions

**Conceptualization:** Marco Geigges, Renato Paro.

**Formal analysis:** Marco Geigges, Geethika Arekatla.

**Funding acquisition:** Marco Geigges, Renato Paro.

**Investigation:** Marco Geigges, Geethika Arekatla.

**Methodology:** Marco Geigges, Renato Paro.

**Project administration:** Marco Geigges.

**Resources:** Marco Geigges, Geethika Arekatla.

**Supervision:** Marco Geigges, Renato Paro.

**Validation:** Marco Geigges.

**Visualization:** Marco Geigges.

**Writing – original draft:** Marco Geigges.

**Writing – review & editing:** Marco Geigges, Geethika Arekatla, Renato Paro.

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
