## [Decision Letter · Decision Letter 0]

31 Dec 2019

PONE-D-19-32589

Effects of priming exposures to lipopolysaccharides on the induction of Polycomb target genes upon re-exposure

PLOS ONE

Dear Prof. Paro,

Thank you for submitting your manuscript to PLOS ONE. After careful consideration, we feel that it has merit but does not fully meet PLOS ONE’s publication criteria as it currently stands. Therefore, we invite you to submit a revised version of the manuscript that addresses the points raised during the review process.

We would appreciate receiving your revised manuscript by Feb 14 2020 11:59PM. To enhance the reproducibility of your results, we recommend that if applicable you deposit your laboratory protocols in protocols.io, where a protocol can be assigned its own identifier (DOI) such that it can be cited independently in the future. For instructions see: http://journals.plos.org/plosone/s/submission-guidelines#loc-laboratory-protocols

We look forward to receiving your revised manuscript.

Kind regards,

Charalampos G Spilianakis, Ph.D

Academic Editor

PLOS ONE

Journal Requirements:

Reviewers' comments:

Reviewer's Responses to Questions

**Comments to the Author**

1. Is the manuscript technically sound, and do the data support the conclusions?

Reviewer #1: Partly

Reviewer #2: Yes

2. Has the statistical analysis been performed appropriately and rigorously? 

Reviewer #1: Yes

Reviewer #2: Yes

3. Have the authors made all data underlying the findings in their manuscript fully available?

Reviewer #1: Yes

Reviewer #2: Yes

4. Is the manuscript presented in an intelligible fashion and written in standard English?

Reviewer #1: No

Reviewer #2: Yes

5. Review Comments to the Authors

Reviewer #1: The manuscript by Geigges et al. follows a clear hypothesis of the epigenetic Polycomb (Pc) gene regulatory system mediating an adaptive innate immune response in Drosophila S2 cells. To test this idea, several target genes of LPS exposure were identified on the basis of previously published studies. Since the authors found the fluorescence detection of responses to LPS stimulus by FISH analysis not sensitive enough, a reporter cell line carrying a GFP downstream of one of the putative Pc regulated genes of innate immune response was generated. In multiple schemes of treatment and exposure to LPS, the authors did not detect an adaptive response of Pc target genes to LPS treatment in S2 cells.

While the overall results of the study are negative, the experimental schemes are generally carefully thought out and executed. The work provides a clear conclusion that can be of value to the scientific community.

Criticism:

The title of the manuscript reflects the experiments but not the results of the work.

I found the abstract confusing and difficult to read. It took me some time to comprehend what the work is about.

The authors use a single reference for RT-PCR quantification and normalization. A second control is mentioned but the results are not standardized to more than one reference. As references can be deceiving, most studies make use of multiple references for RT-PCR analysis and this has become a standard for this kind of experiment.

The authors identify Mmp1 as a suitable reporter for their treatment scheme on the basis of its induction level. A) The induction levels should be specified in table 1 for a reader to have all this information. B) While this rationale seems fair, why were other or better additional targets not considered for constructing reporters? A single reporter might introduce a bias into the study. C) Only the promoter region of Mmp1 was included in the reporter construct. What is the rationale for considering this region a sufficient element for mediating the LPS effect? The authors mention in the discussion that distal gene regulatory elements could be involved in a putative adaptive immune response.

Is it clear that the cells divide normally and are healthy (i.e. not apoptotic) after the LPS stimulus?

Reviewer #2: Geigges et al set out to study the possible memorization of the innate immune response of Drosophila S2 cells to stimulation by bacterial LPS. Suspecting that the PRC epigenetic regulators may play a role in such a memory, they mined pre-existing genomic data for genes induced by LPS and at the same time are positioned near chromatin binding peaks of Pc, Ph and Psc. They studied four such Pc-target/LPS-induced genes, Mmp1, upd2, PGRP-LF, puc, and a non-Pc-target LPS-induced gene CecA by RT-qPCR after LPS stimulation. Disappointingly, the kinetics and amplitude of the transcriptional response of all these genes was not affected by prior exposure to LPS, whether single or multiple. Besides testing the original S2 cell line ("wt S2 cells"), they generated a subclone stably transformed with an Mmp1 promoter-GFP reporter construct to be able to study the population profile of the response to LPS by flow cytometry. Again, this line showed no difference in kinetics, amplitude or population profile of GFP expression whether it had been exposed to LPS before or not. The five genes studied in wt S2 cells were also studied in the reporter subclone and again no effect of prior LPS exposure was seen. Even when challenged with low doses of LPS, that produce a lower level of target gene induction, no differences were observed.

Since the target genes tested showed no sign of transcriptional memory in either of the cell lines, the role of the PRC could not be addressed. So, this paper gives us no insight on whether epigenetic regulators play a role in innate immune memory, since the authors were never able to observe such memory in the first place. The only conclusion that can be reached is that four Pc target immune response genes show no transcriptional memory to earlier immune stimulation, same as the non-Pc-target CecA. Even this conclusion needs one more experiment in order to be solid: The authors should confirm that these four genes are indeed in a Pc-chromatin domain in their new reporter cell line (and that CecA is not). An interesting observation of this paper is that wt S2 cells responded stochastically to LPS stimulation, only 25% of the cells responded. In contrast, in the reporter cell line 80-90% of the cells responded to LPS. Also the amplitude of response was higher in the reporter cell line (compared to wt S2 cells) for 3 out of the 4 Pc-target-genes. Perhaps this subclone has altered chromatin state at these genetic loci (they may have lost the Pc marks) and this may partially explain why no memory was observed. Granted, no memory was detected even in wt S2 cells, but, given the heterogeneity of their response to LPS, we cannot safely conclude that no memory existed on a cell-by-cell basis. Another point that the authors could highlight is that the Pc targets show low levels of transcriptional induction, 2-20x, whereas CecA shows close to 1000x induction upon LPS treatment. Could their association with Pc chromatin dampen their level of induction?

In conclusion, whereas the experiments are well performed, controlled and clearly presented, they fall short of providing any insight on either Pc function or innate immunity memory. This negative result could be worth publishing if the chromatin state of the genes under study is determined in the cell line used, as described above.

6. PLOS authors have the option to publish the peer review history of their article (what does this mean?). If published, this will include your full peer review and any attached files.

Reviewer #1: No

Reviewer #2: No

---

## [Author Response · Author response to Decision Letter 0]

14 Feb 2020

see attached response to reviewers

---

## [Decision Letter · Decision Letter 1]

12 Mar 2020

PONE-D-19-32589R1

Priming exposures to lipopolysaccharides do not affect the induction of Polycomb target genes upon re-exposure

PLOS ONE

Dear Prof. Paro,

Thank you for submitting your manuscript to PLOS ONE. After careful consideration, we feel that it has merit but does not fully meet PLOS ONE’s publication criteria as it currently stands. Therefore, we invite you to submit a revised version of the manuscript that addresses the points raised during the review process.

Please pay attention to Reviewer's #2 comments, especially the way ChIP-qPCR data are presented. 

We would appreciate receiving your revised manuscript by Apr 26 2020 11:59PM. To enhance the reproducibility of your results, we recommend that if applicable you deposit your laboratory protocols in protocols.io, where a protocol can be assigned its own identifier (DOI) such that it can be cited independently in the future. For instructions see: http://journals.plos.org/plosone/s/submission-guidelines#loc-laboratory-protocols

We look forward to receiving your revised manuscript.

Kind regards,

Charalampos G Spilianakis, Ph.D

Academic Editor

PLOS ONE

Reviewers' comments:

Reviewer's Responses to Questions

**Comments to the Author**

1. If the authors have adequately addressed your comments raised in a previous round of review and you feel that this manuscript is now acceptable for publication, you may indicate that here to bypass the “Comments to the Author” section, enter your conflict of interest statement in the “Confidential to Editor” section, and submit your "Accept" recommendation.

Reviewer #1: All comments have been addressed

Reviewer #2: (No Response)

2. Is the manuscript technically sound, and do the data support the conclusions?

Reviewer #1: Yes

Reviewer #2: Yes

3. Has the statistical analysis been performed appropriately and rigorously? 

Reviewer #1: Yes

Reviewer #2: Yes

4. Have the authors made all data underlying the findings in their manuscript fully available?

Reviewer #1: Yes

Reviewer #2: Yes

5. Is the manuscript presented in an intelligible fashion and written in standard English?

Reviewer #1: Yes

Reviewer #2: Yes

6. Review Comments to the Author

Reviewer #1: The manuscript has significantly improved in depth of analysis and clarity of presentation in the revision process. I am overall satisfied in the way the authors have handled my criticism and recommend publication.

Reviewer #2: The authors performed the ChIP experiments needed to confirm the Pc-bound state of two of the genes used in their study. However, the ChIP data presentation is not informative. The authors state (in the supplementary methods) that "results for each target locus were displayed as fold enrichment over negative control." I take this to mean IP(Mmp1)/ IP (non-target region near bx). The correct way to show the results would be to present IP(Mmp1)/input(Mmp1) and as a separate data point to present IP(near bx)/ Input (near bx). This would show the level of enrichment and would avoid the enormous apparent increase in Pc enrichment at the transgene (50x compared to the same region in wt S2 cells), which I strongly believe is simply due to higher copy number of the transgene with respect to the non-target locus in the denominator.

A couple more points that I noticed on rereading the manuscript:

(1) Suppl. Fig. 4: Panels D and E disagree. Based on the histograms in panel D, the mean GFP intensity (panel E) at the two earliest timepoints should show significant differences between pretreated and untreated cells.

(2) Lines 326-330 describe an increased amplitude of response for Mmp1 in the GFP line compared to wt S2 cells. The most logical explanation is that this is simply a population effect, since 90% of the cells respond to LPS in the GFP line, compared to only 25% in the wt S2 (and the RpL32 gene used for normalization is expressed in all cells). This population effect is acknowledged in lines 520-523 in the Discussion. I would add a sentence or two (in the Discussion) to speculate on the other targets that do NOT show similar 4x higher induction levels in the reporter lene vs wt S2 cells. Could it be that the 25% - 90% change in response between wt S2 cells and reporter line is particular to the Mmp1 gene and puc, PGRP-LF etc would show a more equivalent responding population?

7. PLOS authors have the option to publish the peer review history of their article (what does this mean?). If published, this will include your full peer review and any attached files.

Reviewer #1: No

Reviewer #2: No

---

## [Editor Report · Decision Letter 2]

25 Mar 2020

Priming exposures to lipopolysaccharides do not affect the induction of Polycomb target genes upon re-exposure

PONE-D-19-32589R2

Dear Dr. Paro,

We are pleased to inform you that your manuscript has been judged scientifically suitable for publication and will be formally accepted for publication once it complies with all outstanding technical requirements.

With kind regards,

Charalampos G Spilianakis, Ph.D

Academic Editor

PLOS ONE

---

## [Editor Report · Acceptance letter]

27 Mar 2020

PONE-D-19-32589R2 

Priming exposures to lipopolysaccharides do not affect the induction of Polycomb target genes upon re-exposure 

Dear Dr. Paro:

I am pleased to inform you that your manuscript has been deemed suitable for publication in PLOS ONE. Congratulations! Your manuscript is now with our production department. 

With kind regards,

on behalf of

Dr. Charalampos G Spilianakis 

Academic Editor

PLOS ONE